# Novel Cocrystals of Vonoprazan: Machine Learning-Assisted Discovery

**DOI:** 10.3390/pharmaceutics14020429

**Published:** 2022-02-16

**Authors:** Min-Jeong Lee, Ji-Yoon Kim, Paul Kim, In-Seo Lee, Medard E. Mswahili, Young-Seob Jeong, Guang J. Choi

**Affiliations:** 1Department of Pharmaceutical Engineering, Soonchunhyang University, Asan 31538, Chungnam, Korea; mj.sirius1117@gmail.com; 2Department of Medical Science, Soonchunhyang University, Asan 31538, Chungnam, Korea; zlyn97@naver.com (J.-Y.K.); ibanez1994@naver.com (P.K.); dlstj129@naver.com (I.-S.L.); 3Department of ICT Convergence, Soonchunhyang University, Asan 31538, Chungnam, Korea; medardedmund25@sch.ac.kr; 4Department of Computer Engineering, Chungbuk National University, Cheongju 28644, Chungbuk, Korea; ysjay@chungbuk.ac.kr

**Keywords:** pharmaceutical cocrystal, vonoprazan, P-CAB, machine learning, stability

## Abstract

Vonoprazan (VPZ) is the first-in-class potassium-competitive acid blocker (P-CAB), and has many advantages over proton pump inhibitors (PPIs). It is administered as a fumarate salt for the treatment of acid-related diseases, including reflux esophagitis, gastric ulcer, and duodenal ulcer, and for eradication of *Helicobacter pylori*. To discover novel cocrystals of VPZ, we adopted an artificial neural network (ANN)-based machine learning model as a virtual screening tool that can guide selection of the most promising coformers for VPZ cocrystals. Experimental screening by liquid-assisted grinding (LAG) confirmed that 8 of 19 coformers selected by the ANN model were likely to create new solid forms with VPZ. Structurally similar benzenediols and benzenetriols, i.e., catechol (CAT), resorcinol (RES), hydroquinone (HYQ), and pyrogallol (GAL), were used as coformers to obtain phase pure cocrystals with VPZ by reaction crystallization. We successfully prepared and characterized three novel cocrystals: VPZ–RES, VPZ–CAT, and VPZ–GAL. VPZ–RES had the highest solubility among the novel cocrystals studied here, and was even more soluble than the commercially available fumarate salt of VPZ in solution at pH 6.8. In addition, novel VPZ cocrystals had superior stability in aqueous media than VPZ fumarates, demonstrating their potential for improved pharmaceutical performance.

## 1. Introduction

Vonoprazan (VPZ; Takecab^®^) is an orally bioavailable potassium-competitive acid blocker (P-CAB) developed by Takeda Pharmaceutical Company (Osaka, Japan) for the treatment and prevention of acid-related diseases, as the first in a new class of drugs that inhibit gastric H^+^/K^+^-ATPase at the final stage of the acid secretory pathway in gastric parietal cells [1,2]. Compared to conventional proton pump inhibitors (PPIs), P-CAB competes with potassium to inhibit H^+^/K^+^-ATPase and does not require acid-induced activation [3]. While PPIs typically require approximately 3–5 days to exert a maximal effect on gastric acid secretion, P-CABs produce strong and long-lasting inhibitory activity from the first dose [4,5,6].

VPZ is used as a fumarate salt and is approved for use in some countries, including Japan and Korea, for the treatment of acid-related diseases, such as erosive esophagitis, gastric ulcer, duodenal ulcer, peptic ulcer, gastroesophageal reflux, and reflux esophagitis, and for eradication of *Helicobacter pylori* [7]. The lower dose of 10 mg has been approved in Japan for the prevention of duodenal and gastric ulcer recurrence in patients receiving aspirin or nonsteroidal anti-inflammatory drugs (NSAIDs) [8]. In addition, FDA approval is pending for treatment of *H. pylori* infection with VPZ as part of triple therapy in combination with amoxicillin and clarithromycin, or as part of dual therapy in combination with amoxicillin [9].

Over 80% of commercial drugs are solid dosage forms, such as tablets, with oral administration being the preferred drug delivery route. As solid-state formulations, polymorphs, amorphous forms, cocrystals, salts, and their hydrates can be used to alter the physicochemical properties of drugs [10]. The physicochemical properties of an active pharmaceutical ingredient (API), such as stability, dissolution, hygroscopicity, and solubility, have a direct impact on the processing, delivery, and therapeutic performance of drugs [11]. The discovery of better solid forms of an API for clinical use is always desirable [12]. Pharmaceutical cocrystals and salts have been investigated extensively over the past several decades due to their ability to modify the physicochemical properties of APIs without changing their molecular structure [13]. Salt formation, which can increase the solubility and dissolution of APIs, still has the greatest potential [14,15,16]. In terms of stability, however, salts have the drawback that they are more likely to form hydrates compared to cocrystals and many exhibit strong hygroscopicity, making them prone to deliquescence upon moisture sorption [17,18].

Pharmaceutical cocrystals are homogeneous crystalline materials containing at least one API, with one or more coformers in a crystal lattice with a definite stoichiometric ratio [19]. Formulating problematic drugs into cocrystals is a useful strategy to modify APIs for new drug development and product reformulation [20]. Cocrystallization has unique advantages over other solid forms, allowing continued use of old small-molecule drugs that might otherwise disappear due to the introduction of new drugs with better tolerance and efficacy. In addition, as intellectual properties, pharmaceutical cocrystals provide opportunities for inventors to extend the drug lifecycle [11].

The identification of appropriate coformers is the most challenging step in cocrystal development, because the number of coformers that must be screened may run into the hundreds, and more than one technique is often applied to confirm cocrystal formation [21]. Therefore, strategies for rational coformer screening are necessary, and computational screening can guide the selection of a subset of the most promising coformers for an API in subsequent cocrystallization experiments [22]. Various virtual screening tools have been proposed to identify appropriate coformers for APIs based on hydrogen bond propensity [23], molecular electrostatic potential surface [24,25], Hansen solubility parameter [26], conductor-like screening model for real solvents [27], and crystal structure prediction [28].

Recently, data-driven machine learning (ML) methods have been widely used to generate predictive models from available experimental data [29]. ML is a powerful tool for finding relevant patterns in high dimensional data, as it can allow learning from empirical data and use algorithms to simulate linear or nonlinear relationships between material properties and related features. ML can be classified into four types: supervised learning, unsupervised learning, semi-supervised learning, and reinforcement learning. Many ML algorithms use a supervised learning process, in which a set of input data (feature vector, fingerprint, or descriptors; X) and output data (target properties, also called “label”; Y) are required. The goal of the training process is to establish a mapping between the input to the output, Y = f (X), so that discrete targets Y (in case of classification) or target values Y (for continuous quantity data) for unseen data X can be correctly predicted. For example, given a library of cocrystals in which each compound pair has been labeled as cocrystal or none (not observed as cocrystal), a supervised learning algorithm can be used to learn the relationship between molecular features and cocrystal formation, such that new molecular pairs can be predicted to be cocrystal or none.

Various ML models have been applied for cocrystal prediction, including support vector machines (SVMs) [30], multivariate adaptive regression splines [31], random forest (RF) [32], and network-based link-prediction [33]. In our previous work, we developed a virtual screening model based on an artificial neural network (ANN) algorithm for cocrystal prediction [34]. Here, we used the ANN model as a virtual screening tool to discover novel VPZ cocrystals. We evaluated 51 VPZ–coformer pairs and selected 19 coformers for experimental investigation, of which 3 successfully formed pure cocrystals. The cocrystals were characterized by powder X-ray diffraction (PXRD), differential scanning calorimetry (DSC), thermogravimetric analysis (TGA), and Fourier transform infrared (FTIR) spectroscopy. The solubility and intrinsic dissolution rate (IDR) of the novel cocrystals were determined and compared to the commercial fumarate salt of VPZ.

## 2. Materials and Methods

### 2.1. Materials

VPZ free base (>99.6%, Figure 1) and VPZ fumarate (>99.8%) were purchased from Shandong Chuangye Pharmaceutical Technology Co., Ltd. (Shandong, China) and Zhejiang Zetian Fine Chemicals Co., Ltd. (Hangzhou, China), respectively, and used without further purification. The coformers (>97%) listed in Appendix A were purchased from Sigma-Aldrich (St. Louis, MO, USA), Tokyo Chemical Industry (Tokyo, Japan), and Alfa Aesar (Ward Hill, MA, USA) and used as received. High-performance liquid chromatography (HPLC) grade common organic solvents, such as acetonitrile, methanol, and ethanol, were purchased from Honeywell Co. (Morris Plains, NJ, USA) and used as received.

### 2.2. Prediction of Cocrystal Formation of VPZ with 51 Coformers

In a recent study, we developed a classification model for predicting the cocrystal formation of API–coformer pairs [34]. The dataset contained 1476 compound pairs, which were established from an extensive literature review of experimental cocrystal screening involving various APIs. We extracted molecular descriptors from SMILES (Simplified Molecular Input Line Entry System) strings of each compound using the Mordred cheminformatics toolkit. The experimentally observed cocrystal pairs were labeled “1”; the others were labeled “0”. The ANN model outperformed the other ML models when training the model using several ML algorithms, such as SVM, RF, and extreme gradient boost (XGB). It exhibited the best accuracy (0.833), retrieval of the “success” label (0.800), and F1 score (0.817), indicating that the relevant patterns of cocrystal formation were learned based on the collected data.

In the present study, we used the ANN model to predict cocrystal formation. A total of 51 VPZ–coformer combinations were considered, and the molecular descriptors for each compound were generated as described previously. The input of the ANN model consisted of the descriptor values obtained from compound pairs. The model classified each combination as a possible cocrystal (label = 1) or not (label = 0). The prediction results are presented in Appendix A.

### 2.3. Experimental Screening of Cocrystals

Among the 51 coformers evaluated via ML, 19 classified as “1” were experimentally examined for cocrystal formation by liquid-assisted grinding (LAG). Equimolar amounts (0.1 mol) of VPZ and each coformer were deposited into an yttrium-stabilized zirconia mortar, and a few drops of ethanol, acetone, or acetonitrile were added. Grinding was performed manually for 30 min. Otherwise, a Retsch mixer mill (MM-400; Retsch GmbH, Haan, Germany) was used to grind the mixtures of VPZ and each coformer. The mixtures were placed in 10 mL volume stainless steel jars, along with 2 stainless grinding balls of 7 mm diameter and were ground at 25 Hz for 20 min. All trials were conducted at a 1:1 molar ratio.

### 2.4. Bulk Synthesis Using Reaction Crystallization

Bulk synthesis for solubility and dissolution studies was performed by reaction crystallization (RC). A preliminary test for RC was conducted by preparing a saturated solution of the coformer in different solvents, including acetonitrile, acetone, and ethanol, in a sealed vial, with addition of VPZ and stirring until it did not dissolve further. The suspension was stirred for different times, and the resulting solids were filtered and analyzed by PXRD.

Vonoprazan–resorcinol (VPZ–RES): Approximately 1.1 g (10 mmol) of RES were completely dissolved in 2 mL of acetonitrile and 1.73 g (5 mmol) of VPZ free base were added to the solution. The prepared solution was magnetically stirred in tightly sealed glass vials for 24 h at room temperature, and the solids produced were filtered, dried, and collected.

Vonoprazan–catechol (VPZ–CAT): Approximately 0.77 g (7 mmol) of CAT were completely dissolved in 10 mL of acetonitrile and 2.42 g (7 mmol) of VPZ free base were added to the solution in increments of 345 mg (1 mmol). The prepared solution was magnetically stirred in tightly sealed glass vials at room temperature, and crystals were formed within 10 min. After further stirring for 1–2 h, the solids produced were filtered, dried, and collected.

Vonoprazan–pyrogallol (VPZ–GAL): Approximately 1.26 g (10 mmol) of GAL were completely dissolved in 15 mL of acetonitrile and 3.45 g (10 mmol) of VPZ free base were added to the solution in increments of 690 mg (2 mmol). The prepared solution was magnetically stirred in tightly sealed glass vials at room temperature, and crystals were formed within 10 min. After further stirring for 1–2 h, the solids produced were filtered, dried, and collected.

### 2.5. Solid-State Characterization

PXRD patterns were acquired using an X-ray diffractometer (MiniFlex 600; Rigaku, Tokyo, Japan), with a CuKα radiation source (λ = 1.5406 Å) at 40 kV and 15 mA. Each PXRD measurement was conducted over a 2θ range of 4–40°, with a step size of 0.02° and scan rate of 10°/min.

DSC analysis was performed using a DSC-60 calorimeter (Shimadzu, Kyoto, Japan). Each sample (2–3 mg) was placed on an alumina pan (with a blank pan used as a reference) and scanned from 30 to 250 °C at a heating rate of 10 °C/min in a nitrogen atmosphere (N_2_ flow rate: 50 mL/min).

The temperature-dependent weight change was determined using a TGA instrument (N-1000; Scinco, Seoul, Korea). Specifically, 10 mg of each sample was loaded in a platinum holder and heated from 25 to 800 °C at 10 °C/min under nitrogen purging conditions.

A Spectrum Two^®^ attenuated total reflectance (ATR)–FTIR spectrometer (Perkin Elmer, Waltham, MA, USA) was used to acquire infrared (IR) spectra of various specimens in the solid state. Each spectrum was collected in a wavenumber range of 4000–450 cm^−1^ with a resolution of 8 cm^−1^.

Proton nuclear magnetic resonance (^1^H NMR) spectra were recorded on a JEOL ECS 400 MHz NMR spectrometer. Each sample was dissolved in deuterated dimethyl sulfoxide (DMSO-d6) for analysis and chemical shifts for proton are reported in parts per million (ppm) downfield from tetramethylsilane.

### 2.6. Solubility and Intrinsic Dissolution Rate (IDR)

Excess amounts (~500 mg) of powder samples were added to 10 mL of pH 1.2 solution and 30 mL of pH 6.8 solution. The suspension was stirred at 200 rpm in a shaking incubator (SI-600R; JEIO Tech, Daejeon, Korea) at 37 °C for 24 h to measure the equilibrium solubility. After 24 h, aliquots of 2.5 mL were withdrawn from the suspension using a 0.45 μm nylon syringe filter and diluted immediately, followed by analysis of the VPZ concentration by HPLC.

IDR was investigated using a Distek 2100C dissolution tester (Distek, North Brunswick, NJ, USA). Samples of 50 mg of each compound were compressed in a stainless-steel die using a hydraulic press (PIKE Technologies, Inc., Madison, WI, USA) to form drug pellets (d = 13 mm). Only 1 side of the drug pellet was exposed to a solution of 900 mL of phosphate buffer (pH 6.8) in a jar, which was preheated to 37 °C and stirred at 50 rpm. Aliquots of 1.5 mL of the dissolution samples were collected at predetermined time intervals, filtered through 0.45 μm syringe filters, and analyzed for their VPZ concentration by HPLC.

The VPZ concentration was measured using an HPLC system (UFLC; Shimadzu) with a diode array detector and Shim-pack GIS-ODS C18 column (4.6 × 250 mm, 5 μm). HPLC analysis was conducted at a column oven temperature of 40 °C with a flow rate of 1 mL/min, and the mobile phase was a mixture of 20 mM ammonium formate in water (pH 3.05) and HPLC-grade MeOH with the addition of 10 mM triethylamine (TEA) (80% : 20%, *v/v*). The volume of the injected samples was 10 μL and the VPZ content was assayed spectrophotometrically at a wavelength of 245 nm. Calibration plots were constructed over a concentration range of 5–200 μg/mL prior to the solubility and dissolution experiments. An acceptable calibration line (R^2^ = 0.99, n = 5) was obtained.

## 3. Results and Discussion

### 3.1. Virtual and Experimental Screening

Our ANN model was designed to classify a pair of compounds (API and coformer) as a possible cocrystal (label = 1) or not (label = 0). All of the coformers used for predicting the cocrystal formation with VPZ have appeared frequently in pharmaceutical cocrystal research. The full list of coformers (N = 51) investigated is shown in Appendix A. They have different functional groups, including carboxylic acid, amide, hydroxyl, amine, etc., and the majority of the coformers contain carboxylic acids, which have been widely used for cocrystal formation with various APIs. Indeed, carboxylic acid···pyridine synthon is one of the most favored hydrogen bonds in numerous cocrystals [35].

The prediction results showed that 19 of the 51 coformers could form cocrystals with VPZ (Appendix A). Among them, five were phenolic, seven were acidic, and seven were basic compounds. Interestingly, only 6 of 30 molecules with a carboxylic acid group were classified as coformer candidates for VPZ cocrystals. This is because even if these molecules have acidic properties, they all differ in terms of the carbon chain length, molecular complexity, weight, volume, flexibility, etc. [36], and their descriptor values (used as input data for the ANN model) were significantly different. To discover new cocrystals of VPZ, experimental screening with the 19 coformers was carried out using LAG.

PXRD analysis was the primary tool used to verify whether new solid forms were synthesized. All of the solids obtained from LAG were initially characterized by PXRD and are shown in Appendix A, along with comparisons of the respective starting materials. A distinct difference in the PXRD patterns between the products and raw materials confirmed the formation of new crystalline phases. The list of coformers used, and the results of LAG, are summarized in Table 1. Interestingly, none of the seven basic compounds screened formed cocrystals with VPZ. Similarly, LAG of VPZ with 4-hydroxybenzoic acid, pyrrole-2-carboxylic acid, oxamic acid, and saccharin resulted in physical mixtures of the two components. On the other hand, new PXRD patterns were observed in the LAG results for the phenolic and some carboxylic acid coformers (catechol (CAT), hydroquinone (HYQ), resorcinol (RES), pyrogallol (GAL), succinic acid (SUC), and pyroglutamic acid (PGA)), indicating the formation of new crystals. In the cases of methyl-hydroquinone (MHQ) and benzoic acid (BZA), the PXRD patterns of the LAG samples showed an amorphous halo.

As VPZ has a strong basic group (amine) with a *pK_a_* value of 9.4 [37], it is likely to form salts with carboxylic acids. In fact, the formation of a salt or cocrystal can be assessed based on the “Δ*pK_a_* rule”, which states that a salt is formed when Δ*pK_a_* [*pK_a_* (base) − *pK_a_* (acid)] ≥ 3 and a cocrystal is expected when Δ*pK_a_* ≤ 0. The combination with a value 0 ≤ Δ*pK_a_* ≤ 3 is much less predictable and falls within a “salt-cocrystal continuum” [38]. We applied this approach to the coformers selected based on virtual cocrystal screening to assess whether the combination would form a cocrystal or salt (Table 1). From the results, we expected that the LAG products of VPZ with SUC and PGA would be salts, as VPZ–SUC and VPZ–PGA had Δ*pK_a_* values > 3. In contrast, coformers with phenol groups showed Δ*pK_a_* < 0, indicating a higher probability of cocrystal formation with VPZ. Therefore, we judged that the new PXRD patterns of VPZ–CAT, VPZ–HYQ, VPZ–RES, and VPZ–GAL resulted from the formation of cocrystals. As the discovery of new cocrystals for VPZ was the main objective of this study, we focused on these four cocrystals and performed detailed characterization.

### 3.2. Characterization of VPZ Cocrystals

The bulk cocrystals that crystallized using the RC method were characterized by PXRD, DSC, TGA, and FT-IR. The PXRD patterns of the cocrystals prepared by RC showed several distinct diffraction peaks, which were different from the characteristic peaks corresponding to VPZ and the coformers (Figure 2). The absence of characteristic peaks of VPZ and coformers confirmed the high purity of the prepared cocrystals. Comparison of the PXRD patterns of the cocrystals obtained by RC with those of the LAG products showed a good match in all cases, except VPZ–HYQ. In the case of VPZ–HYQ, the results indicated the potential presence of VPZ–HYQ polymorphs, which may have resulted from the use of different preparation methods. In addition, the diffraction patterns of the cocrystals produced by RC exhibited relatively high crystallinity compared to their LAG results (Figure 2). 

The thermal stability and degradation of the VPZ cocrystals were determined by DSC and TGA. Figure 3 compares the thermograms of the starting materials and cocrystals. As can be seen in the figure, VPZ free base exhibited relatively low thermal stability, with a low melting point of 67.3 °C (T_onset_). This may be one of the reasons why the fumarate salt of VPZ is used commercially. The newly discovered VPZ–RES, VPZ–CAT, and VPZ–GAL cocrystals all showed a single endothermic peak corresponding to the melting (T_onset_) of pure cocrystals at 93, 103.9, and 130 °C, respectively. Furthermore, no significant weight change was observed before decomposition, confirming the unsolvated nature of VPZ cocrystals (Appendix A). It is worth noting that the melting points of all three cocrystals were significantly higher than that of VPZ, indicating relatively higher thermal stability of the new phases than VPZ. The enthalpy of fusion values of the 3 cocrystal systems (VPZ–RES: 38 KJ/mol; VPZ–CAT: 53 KJ/mol; VPZ–GAL: 57 KJ/mol) were higher than that of VPZ (27 KJ/mol), and the respective coformers (RES: 23 KJ/mol; CAT: 19 KJ/mol; GAL: 19 KJ/mol), suggesting that the crystal lattice was strengthened upon cocrystallization [39]. Melting behavior can be used to describe the energy of the crystal lattice, and for calculating the aqueous solubility of crystalline organic molecules [40]. A lower melting point promotes aqueous solubility. For VPZ–RES, the melting point was lower than those of VPZ–CAT and VPZ–GAL. Along with the lower enthalpy of fusion, this suggested weaker intermolecular interactions than in the two other VPZ cocrystals. Therefore, it is likely that VPZ–RES will have higher solubility than VPZ–CAT and VPZ–GAL.

For VPZ–HYQ, 2 small endothermic events can be seen in the DSC thermogram, at around 60 and 85 °C, before melting at 99.6 °C (Figure 3d). The first endotherm seemed to be related to the release of solvent and the second may have been attributable to polymorphic phase transition, if the two polymorphs are enantiotropically related according to the heat of transition rule [41]. In the case of the VPZ–HYQ cocrystal system, therefore, further studies are necessary to identify the thermodynamic relationships of the cocrystal polymorphs and produce a pure polymorphic form.

Unfortunately, all attempts to grow crystals of VPZ-RES, VPZ-CAT, and VPZ-GAL suitable for single crystal XRD proved unsuccessful and the cocrystallization experiments always resulted only in microcrystalline samples. However, the stoichiometric ratio of VPZ-RES, VPZ-CAT, and VPZ-GAL was confirmed as 1:1 by ^1^H NMR analysis (Appendix A).

The ATR-FTIR spectra of the starting materials and their cocrystals were recorded in the solid state. Compared to their respective components, the cocrystals showed IR spectral peak shifts in both the fingerprint and high wavenumber regions (Appendix A). The peak shifts for various functional groups indicate the alteration of intermolecular interactions around these groups, and the formation of new solid forms. The peak positions and assignments of the selected peaks of interest are shown in Table 2 [42,43].

The major differences between the cocrystals and their respective components appeared in the region of 3600–2800 cm^−1^ (Figure 4). RES, CAT, and GAL exhibited broad and conspicuous absorption peaks in this region, which corresponded to phenolic O–H stretching (RES: 3188 cm^−1^; CAT: 3444 and 3320 cm^−1^; GAL: 3534 and 3229 cm^−1^). In the VPZ cocrystals, the broad phenolic O–H stretching peaks were shifted to wavenumbers about 60–200 cm^−1^ lower and superimposed on the C–H stretching regions (around 3000 cm^−1^). These marked downward shifts in the region of phenolic O–H stretching suggested that the O–H groups were involved in strong intermolecular hydrogen bonding without proton transfer, thereby forming new phases. The weak N–H stretching band at 3321 cm^−1^ in the VPZ spectrum was also shifted to a slightly lower wavenumber (by around 20 cm^−1^) in the spectra of VPZ cocrystals, indicating the presence of interactions between VPZ and the corresponding coformer molecules.

Interestingly, several new bands appeared in the region of 1600–1000 cm^−1^ in the spectra of VPZ cocrystals (Figure 5). The peak at 1573 cm^−1^ in VPZ was assigned to pyridine C=N stretching, which appeared in the range of 1640–1557 cm^−1^ [40]. In VPZ cocrystals, the characteristic peak of C=N stretching was split into 2 peaks at 1585–1578 and 1568–1563 cm^−1^. A similar peak split was also observed in the range of 1330–1300 cm^−1^. The peak at 1313 cm^−1^ in VPZ was assigned to the aromatic amine C–N stretching, which appeared in the range of 1342–1266 cm^−1^ [44]. In VPZ cocrystals, the C–N stretching split into 2 peaks at 1332–1325 and 1306–1302 cm^−1^. These spectral changes suggested that the pyridine N of VPZ participates in hydrogen bonding based on hydroxy–pyridine synthon when cocrystals are formed. Furthermore, 2 new peaks emerged at 1215 and 1355 cm^−1^ in the spectra of VPZ–CAT and VPZ–GAL, respectively. Again, the appearance of new absorption peaks and a significant shift in the characteristic peaks support the formation of VPZ cocrystals due to alteration of the molecular environment and intermolecular interactions.

### 3.3. Solubility and IDR

The solubility and dissolution behavior of drugs in aqueous solvent are important physicochemical properties for good oral bioavailability. VPZ has pH-dependent solubility, which increases with decreasing pH. The fumarate salt of VPZ is used commercially and is moderately soluble in water with a reported solubility of <1 mg/mL [45]. The aqueous solubility for cocrystals and salts of VPZ was measured in pH 1.2 and 6.8 buffer solutions at 37 °C. VPZ concentrations were determined in aliquots retrieved from the suspensions after 24 h by HPLC. One of the major challenges in investigations of solubility in multicomponent solid forms is the change in the solution composition due to precipitation of the components dissociated over the test period [25]. In this study, the stability of the cocrystals in aqueous solution was checked by PXRD analysis of the solids remaining after solubility experiments. At pH 1.2, however, the amount recovered from the suspensions was too low to confirm their stability. The results of the PXRD analysis of the undissolved residue suggested that the cocrystals were stable at pH 6.8 (Figure 6a–c); therefore, the solubility values measured here can be considered as the true solubilities of the cocrystals. Remarkably, the results of the PXRD analysis of VPZ fumarate salt after slurrying for 24 h in solution at pH 6.8 were different from those of the starting material (Figure 6d), confirming the higher stability of the VPZ cocrystals than commercial VPZ fumarate. The solubility results are presented in Figure 7 and Table 3. Under both conditions, the solubility order was VPZ–RES > VPZ–CAT > VPZ–GAL. This, along with the lower enthalpy of fusion, suggested weaker intermolecular interactions in VPZ–RES than VPZ–CAT and VPZ–GAL. Notably, the solubility values for VPZ–RES were higher than, and similar to, the commercial fumarate salt of VPZ at pH 6.8 and 1.2, respectively.

IDR is a kinetic parameter that is useful for estimating solubility improvement in cocrystals undergoing phase transformation during dissolution. Intrinsic dissolution experiments were carried out at pH 6.8 using a dissolution tester. Samples collected at regular intervals were quantified for their VPZ concentration, and the cumulative concentrations of VPZ are plotted in Figure 8. The IDR values are shown in Table 3. The commercial fumarate salt of VPZ showed the fastest rate of dissolution, followed by VPZ–CAT, VPZ–RES, and VPZ–GAL cocrystals. Remarkably, the dissolution rates of VPZ–CAT and VPZ–RES were only slightly lower than that of the commercial VPZ fumarate. Our results for cocrystals of VPZ demonstrated that cocrystals of APIs can exhibit comparable dissolution rates to salt forms. The IDR plots showed that VPZ fumarate salts reached peak concentrations within 60 min of drug dissolution while the VPZ cocrystals exhibited a gradual increase in the amount of VPZ dissolved for up to 2 h. Taking into consideration the difference in the PXRD results for VPZ fumarate salt after slurrying for 24 h in solution at pH 6.8 compared to those for the starting material, the IDR curves indicated that phase transformation of VPZ fumarate may occur after 1 h. Therefore, our data suggest that VPZ cocrystals could be potential alternatives to the commercial fumarate salt of VPZ.

## 4. Conclusions

In summary, we discovered novel cocrystals of vonoprazan (VPZ) via a combination of virtual and experimental screening. Among the eight new solid forms synthesized, three materials were proved to be the novel VPZ cocrystals, VPZ–RES, VPZ–CAT, and VPZ–GAL, which were eventually characterized. The dissolution rates and solubilities of the VPZ cocrystals were comparable to those of the commercial VPZ fumarate salt. The dissolution behavior of the VPZ cocrystals appears to be mainly determined by the intermolecular interaction as a correlation between the melting point, the enthalpy of fusion, and the dissolution rate exists.

Three VPZ cocrystals and the fumarate salt remained stable in the solid state during the accelerated stability testing. On the other hand, the stabilities of the 3 VPZ cocrystals in pH 6.8 buffer solution were superior to the fumarate salt, suggesting the new VPZ cocrystals as a potential alternative to the marketed VPZ fumarate salt. This study not only contributes to extending the knowledge about the solid state of pharmaceutically important drug substances, but also provides a good example of the successful integration of a machine-learning approach with experimental screening for the discovery of novel pharmaceutical solid forms.

## Figures and Tables

**Figure 1 pharmaceutics-14-00429-f001:**
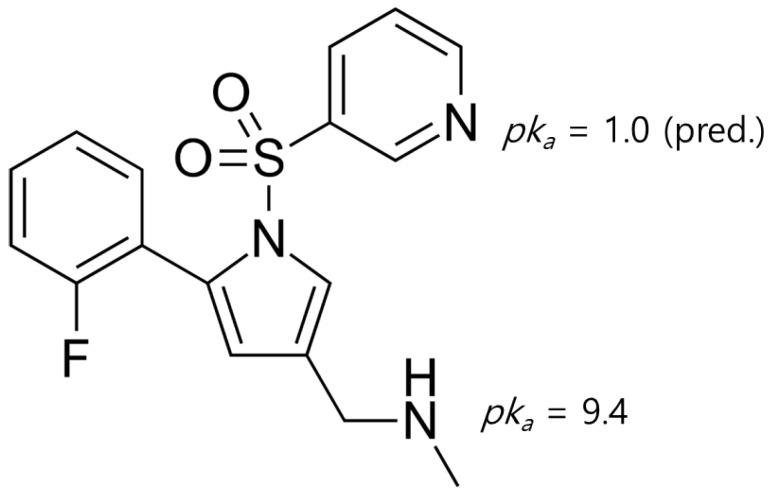
Chemical structure of vonoprazan (VPZ) and *pK_a_* values.

**Figure 2 pharmaceutics-14-00429-f002:**
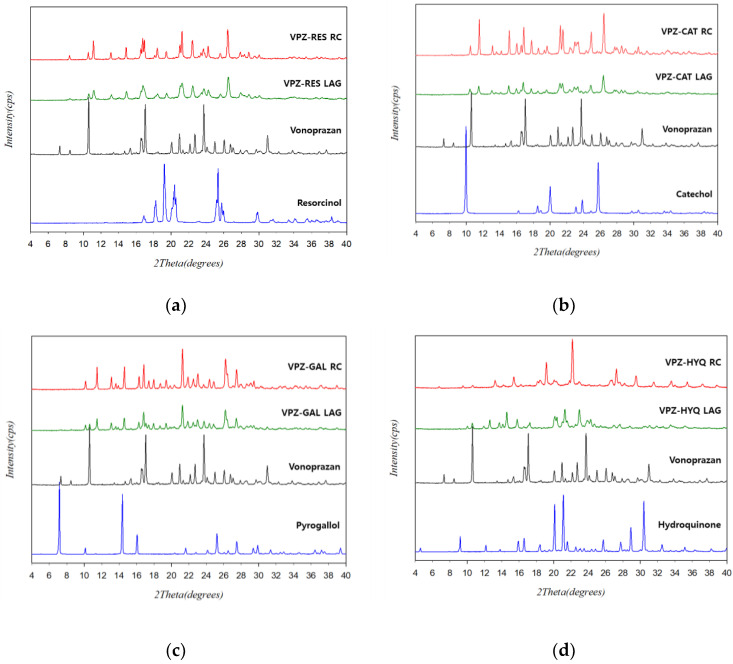
Comparison of the PXRD patterns of (**a**) VPZ-RES, (**b**) VPZ-CAT, (**c**) VPZ-GAL, and (**d**) VPZ-HYQ prepared by RC and LAG with their starting materials.

**Figure 3 pharmaceutics-14-00429-f003:**
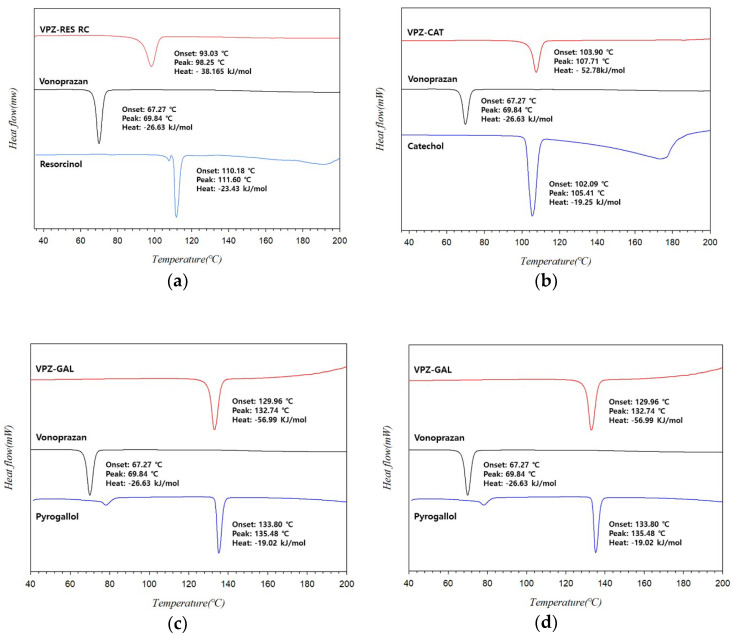
Comparison of the DSC thermograms of (**a**) VPZ-RES, (**b**) VPZ-CAT, (**c**) VPZ-GAL, and (**d**) VPZ- HYQ prepared by RC with their starting materials.

**Figure 4 pharmaceutics-14-00429-f004:**
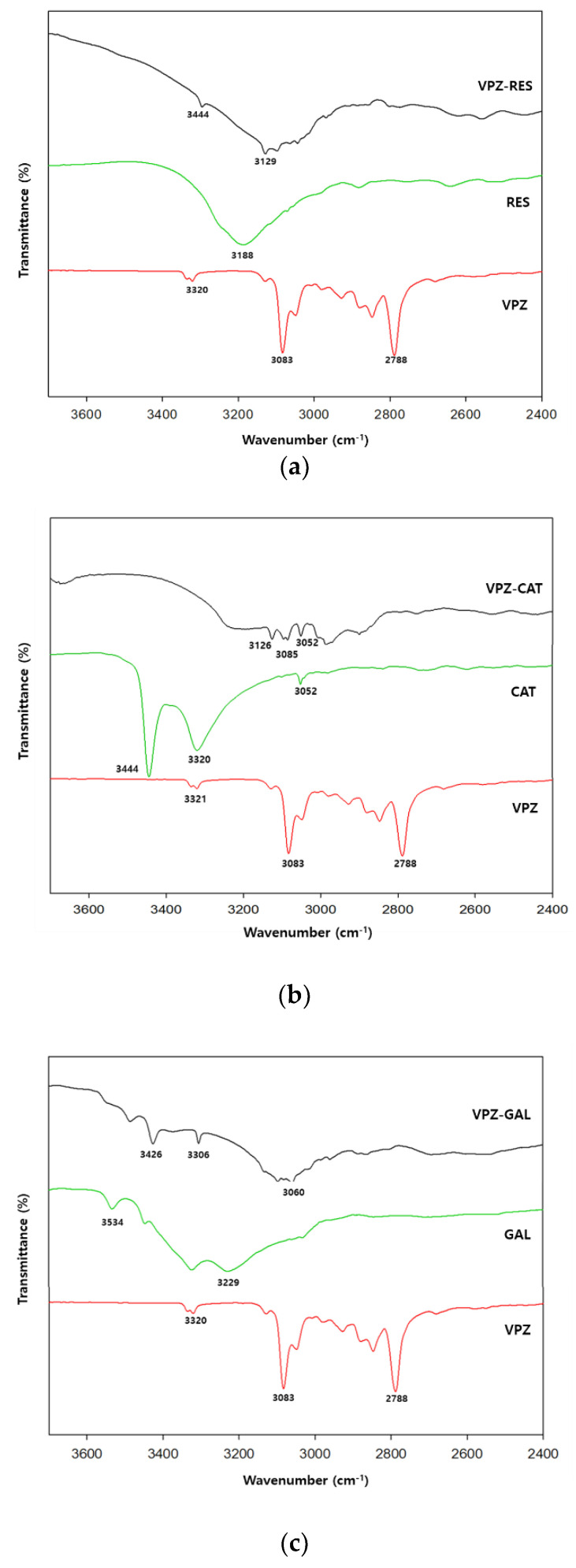
ATR-FTIR spectra of (**a**) VPZ-RES, (**b**) VPZ-CAT, (**c**) VPZ-GAL, and their respective components in the region of 3700–2400 cm−1.

**Figure 5 pharmaceutics-14-00429-f005:**
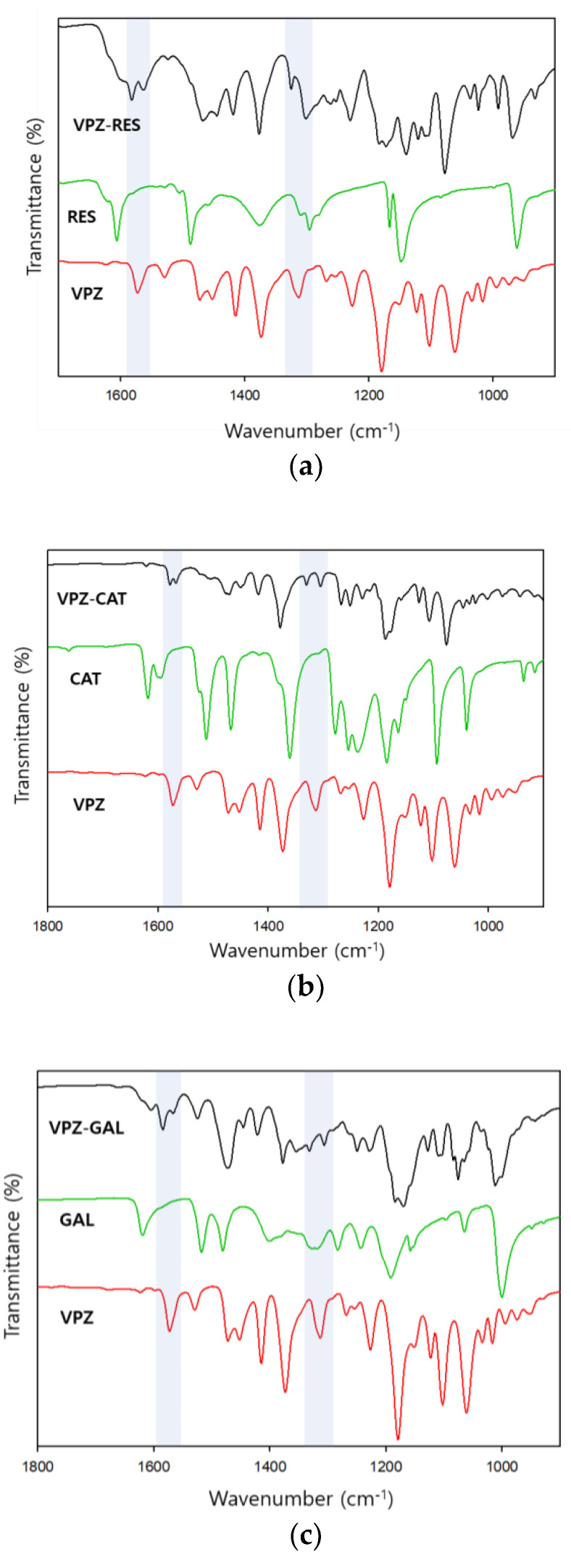
ATR-FTIR spectra of (**a**) VPZ-RES, (**b**) VPZ-CAT, (**c**) VPZ-GAL, and their respective components in the region of 1800–900 cm−1.

**Figure 6 pharmaceutics-14-00429-f006:**
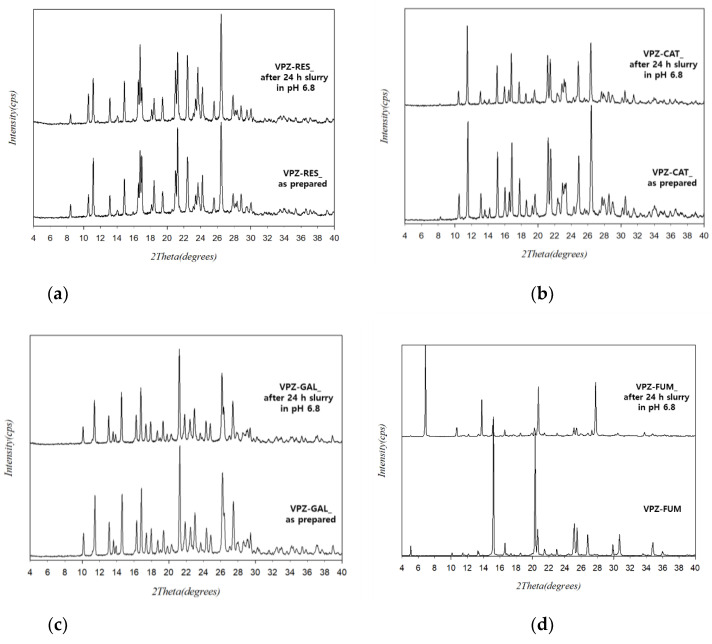
Comparison of the PXRD patterns of (**a**) VPZ-RES, (**b**) VPZ-CAT, (**c**) VPZ-GAL, and (**d**) VPZ fumarate after 24 h slurry.

**Figure 7 pharmaceutics-14-00429-f007:**
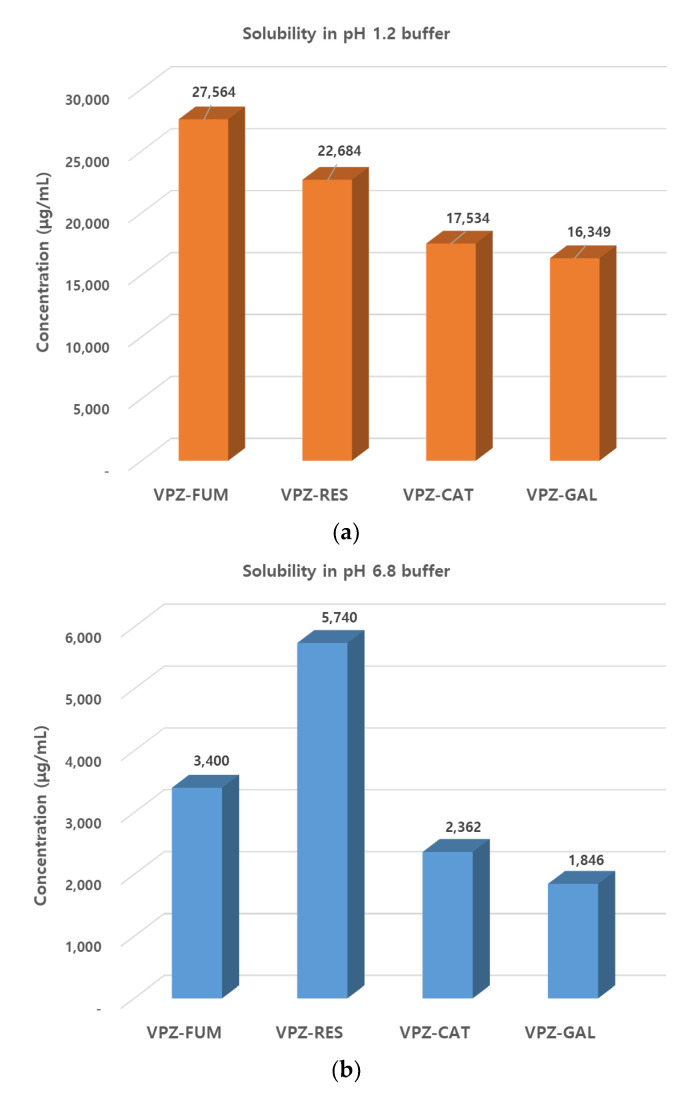
Comparative solubility of VPZ cocrystals and VPZ fumarate in (**a**) pH 1.2 and (**b**) pH 6.8 buffer solution.

**Figure 8 pharmaceutics-14-00429-f008:**
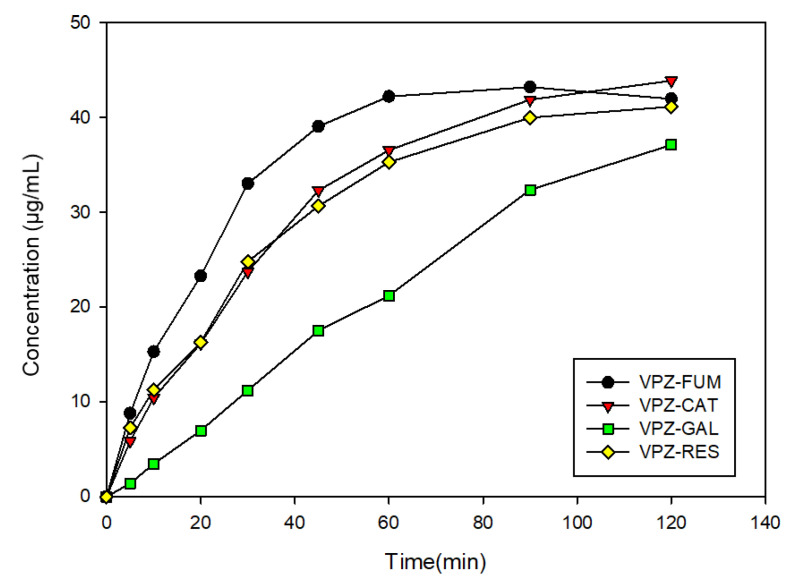
IDR curves of 3 VPZ cocrystals and VPZ fumarate at pH 6.8 in buffer solution.

**Table 1 pharmaceutics-14-00429-t001:** Summary of the coformers used for the formation of cocrystals with VPZ (*pK_a_* = 9.4).

Group	Coformer	LAG Result ^a^	*pK_a_* ^b^	Δ*pK_a_*	Prediction
	Pyrocatechol	√	9.64	−0.24	cocrystal
	Hydroquinone	√	9.98	−0.58	cocrystal
Phenols	Resorcinol	√	9.57	−0.17	cocrystal
	Pyrogallol	√	9.1	0.3	continuum
	Methyl-hydroquinone	–	10.1	−0.7	cocrystal
Acids	Benzoic acid Succinic acid Pyroglutamic acid 4-hydroxybenzoic acid Pyrrole-2-carboxylic acid Oxamic acid saccharin	– √ √ x x x x	4.08 3.86 3.61 4.38 3.62 2.48 1.94	5.32 5.54 5.79 5.02 5.78 6.92 7.46	salt salt salt salt salt salt salt
Amide /Amine	Acetamide Glycolamide Benzamide Nicotinamide Isonicotinamide Urea	x x x x x x	16.75 13.65 14.56 13.39 13.71 16.3	−7.35 −4.25 −5.16 −3.99 −4.31 −6.9	cocrystal cocrystal cocrystal cocrystal cocrystal cocrystal
	Piperazine	x	9.26 (base)	-	-

a √: new phase, x: physical mixture, –: amorphous observed with PXRD. b *pK_a_* were calculated using a ChemAxon calculator.

**Table 2 pharmaceutics-14-00429-t002:** Infrared characteristics of VPZ, coformers, and associated cocrystals.

Wavenumber (cm^−1^)							Peak Assignment [40,42,43,44]
VPZ	RES	VPZ- RES	CAT	VPZ- CAT	GAL	VPZ- GAL	
3321		3296				3306	N-H stretching
	3188	3129	3444 3320	3126	3534 3229	3426 3060	O-H stretching
1623	1606		1618	1622	1620	1605	C=C stretching
1573		1582 1563		1578 1568		1585 1567	C=N stretching
1374 1179		1377 1183		1378 1177		1378 1184	S=O stretching
1313		1325 1302		1330 1305		1332 1306	Aromatic C-N stretching

**Table 3 pharmaceutics-14-00429-t003:** Solubility and dissolution rate of VPZ cocrystals and VPZ fumarate.

Solid Form	Equilibrium Solubility after 24 h in pH 1.2 (mg/mL)	Equilibrium Solubility after 24 h in pH 6.8 (mg/mL)	IDR in pH 6.8 (mg/cm^2^∙min)
VPZ-FUM	27.71 ±0.21	3.4 ±0.45	1.35
VPZ-RES	17.53 ±0.49	5.74 ±0.13	0.99
VPZ-CAT	17.42 ±0.46	2.09 ±0.87	0.9
VPZ-GAL	22.68 ±0.5	1.85 ±0.31	0.69

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
