# Peer review of "Novel Cocrystals of Vonoprazan: Machine Learning-Assisted Discovery"

_pharmaceutics, 2022, doi:10.3390/pharmaceutics14020429_

Round 1

Reviewer 1 Report

The major novelty of this paper is to use machine learning in cocrystal prediction. For the parts in cocrystal preparation and characterization, it can be regarded as a routine solid-state screening programme. Therefore, it is advised that the authors can consider to expand the part in machine learning so that the audiences can be educated in this more. Since the audiences of this journal usually do not have AI/ML background, it will be nice to know more in this area. Thank you. 

Author Response

Manuscript title: Novel Cocrystals of Vonoprazan: Machine Learning-assisted Discovery

Manuscript number: Pharmaceutics-1576531

Corresponding author: Guang J. Choi

===================================================

Referee 1

The major novelty of this paper is to use machine learning in cocrystal prediction. For the parts in cocrystal preparation and characterization, it can be regarded as a routine solid-state screening programme. Therefore, it is advised that the authors can consider to expand the part in machine learning so that the audiences can be educated in this more. Since the audiences of this journal usually do not have AI/ML background, it will be nice to know more in this area. Thank you. 

[A] Thank you very much for your kind advice. We revised the introduction part as below primarily to reinforce the machine learning-related contents in this manuscript:

[Before revision]

Recently, data-driven machine learning (ML) methods have been widely used to generate predictive empirical models from available experimental data [30].

[After revision]

Recently, data-driven machine learning (ML) methods have been widely used to generate predictive models from available experimental data [30]. ML is a powerful tool for finding relevant patterns in high dimensional data, it can allow to learn from empirical data and use algorithms to simulate linear or nonlinear relationships between material properties and related features. ML can be classified to four types, supervised learning, unsupervised learning, semi-supervised learning, and reinforcement learning. Many ML algorithms use a supervised learning process, in which a set of input data (feature vector, fingerprint or descriptors; X) and output data (target properties, also called “label”; Y) are required. The goal of the training process is to establish a mapping be-tween the input to the output, Y = f (X), so that discrete targets Y (in case of classification) or target values Y (for continuous quantity data) for unseen data X can be correctly predicted. For example, given a library of cocrystals in which each compound pair has been labeled as cocrystal or none (not observed as cocrystal), a supervised learning algorithm can be used to learn the relationship between molecular features and cocrystal formation, such that new molecular pairs can be predicted to be cocrystal or none.

Thank you very much again for your great concern.

Reviewer 2 Report

+

The manuscript represents a major effort to find advantageous co-crystal alternatives to Vonoprazan-fumarate.

In general, from a pharmaceutical point of view the strategy for selecting most advantageous co-crystals is well presented.

My main queries are two.

(1) The absence of at least one crystallographic result obtained by single-crystal X-ray diffraction method.

(2) The way in which the conclusions are witted. They are not 'another abstract'! Concluding remarks should emphasize main contribution(s) of the work.

Minor points.

(a) The assignments of a good part of the bands in the FT-IR spectra seem debatable to me. I honestly think that an expert in this type of spectroscopy does not carry them out.

(b) Note that References 26 and 47 correspond to the same paper.

Author Response

Referee 2

The manuscript represents a major effort to find advantageous co-crystal alternatives to Vonoprazan-fumarate. In general, from a pharmaceutical point of view the strategy for selecting most advantageous co-crystals is well presented.

Thank you very much for your kind comments.

My main queries are two.

(1) The absence of at least one crystallographic result obtained by single-crystal X-ray diffraction method.

[A] If we had been successful in determining the crystal structure of at least one VPZ cocrystal by single-crystal XRD analysis, we should have included the SC-XRD results so that this manuscript would have been much better. As a matter of fact, we have given considerable efforts to obtain single crystals of three cocrystals in a sufficient size for over a whole year. Unfortunately, however, we were not able to get big-sized crystals suitable for SC-XRD analysis even with some experts’ advices and the newly purchased apparatus to make single crystal. Accordingly, we focused on the other characterization of three new cocrystals in this manuscript, instead. However, we are now continuing to work on the preparation of single crystals. So, we would be ready to discuss the crystal structures of the VPZ cocrystals later.

(2) The way in which the conclusions are witted. They are not 'another abstract'! Concluding remarks should emphasize main contribution(s) of the work.

[A] The conclusion was revised as below:

[Before revision]

We discovered novel cocrystals of VPZ via a combination of virtual and experimental screening. Our ANN model designated 19 of 51 molecules as potential coformers for VPZ cocrystals. Among them, eight candidates were considered as likely to produce new solid forms through LAG. Three novel VPZ cocrystals, VPZ–RES, VPZ–CAT, and VPZ–GAL, were eventually identified and characterized. The dissolution rates and solubilities of the VPZ cocrystals were comparable to those of the commercial VPZ fumarate salt. On the other hand, the stabilities of the three VPZ cocrystals in pH 6.8 buffer solution were superior to the VPZ fumarate salt. Therefore, the VPZ cocrystals discovered with ML can provide a means to improve the performance of currently available commercial VPZ fumarate salt drugs.

[After revision]

In summary, we have discovered novel cocrystals of vonoprazan (VPZ) via a combination of virtual and experimental screening. Among eight new solid forms synthesized, three materials were proved to be the novel VPZ cocrystals, VPZ–RES, VPZ–CAT, and VPZ–GAL, which were eventually characterized. The dissolution rates and solubilities of the VPZ cocrystals were comparable to those of the commercial VPZ fumarate salt. The dissolution behavior of the VPZ cocrystals appears to be mainly determined by the intermolecular interaction as there exists a correlation between the melting point, the enthalpy of fusion, and the dissolution rate.

Three VPZ cocrystals as well as the fumarate salt remained stable in solid state during the accelerated stability testing. On the other hand, the stabilities of the three VPZ cocrystals in pH 6.8 buffer solution were superior to the fumarate salt suggesting the new VPZ cocrystals as a potential alternative to the marketed VPZ fumarate salt. This study not only contributes to extend the knowledge about the solid state of pharmaceutically important drug substances, but provides a good example of the successful integration of a machine-learning approach with an experimental screening for the discovery of novel pharmaceutical solid forms.

----------------------------------------------------------------------------------------

Minor points.

(a) The assignments of a good part of the bands in the FT-IR spectra seem debatable to me. I honestly think that an expert in this type of spectroscopy does not carry them out.

[A] We were feeling the same thing. In the cases of three coformers (RES, CAT, and GAL), we confirmed that their IR spectra are in a good agreement with published data. However, no references for IR spectrum of VPZ were found in the literature so far. So, we have tried to interpret the IR data of VPZ by employing the IR spectra of compounds similar to the VPZ, which seems to be the best approach at this moment. In addition, a possible argument regarding the IR band interpretation might not be the main point of this study.

(b) Note that References 26 and 47 correspond to the same paper.

[A] A correction has been made by deleting the reference # 47.

Thank you very much again for your great concern.